# Association between Body Mass Index and Diabetes Mellitus Are Mediated through Endogenous Serum Sex Hormones among Menopause Transition Women: A Longitudinal Cohort Study

**DOI:** 10.3390/ijerph20031831

**Published:** 2023-01-19

**Authors:** Li He, Bingbing Fan, Chunxia Li, Yanlin Qu, Ying Liu, Tao Zhang

**Affiliations:** 1Department of Biostatistics, School of Public Health, Cheeloo College of Medicine, Shandong University, Jinan 250012, China; 2Institute for Medical Dataology, Shandong University, Jinan 250002, China

**Keywords:** mediation analysis, sex hormones, perimenopause, overweight, diabetes, menopause women, longitudinal study

## Abstract

Objective: To explore whether and to what extent endogenous sex hormones mediate the association between overweight and diabetes risk in menopausal transition women. Methods: Premenopausal women were from the Study of Women’s Health Across the Nation, with measurements of serum sex hormone including sex hormone binding globulin (SHBG), testosterone (T), estradiol (E2), follicle-stimulating hormone (FSH), and dehydroepiandrosterone sulfate (DHAS) in first postmenopausal follow-up. At the last postmenopausal follow-up, hyperglycemia status was confirmed. The partial least squares (PLS) regression method was used to extract hormonal signals associated with body mass index (BMI). Hyperglycemia was defined as individuals with prediabetes or diabetes; overweight was defined as BMI ≥ 25 kg/m^2^. Causal mediation analysis was used to examine the mediation effect on the association between perimenopause overweight and post-menopause hyperglycemia through PLS score and individual sex hormones. Results: The longitudinal study included 1438 normal glucose women with a baseline mean age (SD) of 46.5 (2.6) years and a mean follow-up period of 9.9 years. During the follow-up period, 145 (10.1) cases of hyperglycemia occurred. Compared with normal-weight participants, overweight women were associated with a higher hyperglycemia risk during the transition period (OR = 4.06, 95% CI: 2.52 to 6.80). Overweight women had higher T, E2, and lower SHBG, FSH, and DAHS concentrations (β = 0.26, 0.38, −0.52, −0.52, and −0.13, *p* < 0.05 for all). After adjusting for overweight and covariates, lower SHBG and FSH levels were associated with higher hyperglycemia risk (OR = 0.70 and 0.69, all *p* < 0.05). As a linear combination of sex hormones, the PLS score was positively associated with T, E2, and negatively with SHBG, FSH, and DHAS. PLS score interpreted 36.50% (*p* < 0.001) of the overweight-hyperglycemia association. Considering single-sex hormones, the mediation proportion of SHBG and FSH were 21.38% (*p* < 0.001) and 24.08% (*p* < 0.001). Conclusions: Sex hormones mediated the association of overweight and diabetes risk in menopause transition women. SHBG and FSH have the dominant mediation effect.

## 1. Introduction

Diabetes is a major public health challenge worldwide, as it increases the risk of mortality and adverse cardiovascular events [1]. Being overweight or obese is the most significant risk factor for the development of diabetes, especially in women. Studies have shown that aberrant estrogenic signaling and elevated levels of androgens are among some of the proposed mechanisms explaining the heightened diabetes risk in women [2,3]. It is worth mentioning that the prevalence of diabetes is lower in premenopausal women, whereas it increases significantly after the age of 45–54 years, which is partly due to the weakened protective effect of female hormones. Menopause is a specific journey for women who will face major changes in endocrine hormones and increased diabetes risk, and understanding the mechanisms underlying overweight-associated menopausal diabetes will help discover novel prevention strategies for diabetes.

We hypothesized that endogenous hormonal changes could partially explain the association between overweight/obesity and diabetes in the menopausal transition period. Numerous studies have confirmed the significant differences in hormone levels between obese and normal women [4,5,6,7,8,9,10,11]. In an all-female cohort study, obesity was associated with lower concentrations of sex hormone binding protein (SHBG), estradiol (E2), and higher concentrations of testosterone (T) [12]. A meta-analysis of six RCTs showed that weight loss interventions in healthy postmenopausal women for 16–52 weeks were effective in altering multiple sex hormone concentrations [13]. Similarly, hormonal changes during perimenopause or menopausal transition were found to increase the risk of diabetes in women [14,15,16,17]. In three population-wide prospective cohorts, it was consistently observed that lower plasma SHBG concentrations and higher T concentrations were associated with a lower risk of diabetes in women [14,18]. In addition, a study of postmenopausal women indicated that obesity and androgens were independently associated with insulin sensitivity [19]. However, whether obesity or being overweight in women during the menopause period affects the onset of diabetes through endogenous sex hormone changes was unclear or accurately quantified.

Therefore, this study aims to explore whether E2, T, FSH, DHAS, and SHBG mediate the association between overweight/obesity and diabetes during the menopause transition period within the Study of Women’s Health Across the Nation (SWAN) cohort. Considering the complex relationships among sex hormones, we calculate a score from partial least squares (PLS) regression analyses to represent compound hormonal variation characteristics and examine its mediation effect on the association between overweight/obesity and diabetes.

## 2. Materials and Methods

### 2.1. Study Population

The Study of Women’s Health Across the Nation (SWAN) cohort, a multicenter, multiracial longitudinal study, was designed to describe physical and psychological changes in community-based perimenopausal women. The SWAN cohort was sampled at the following 7 sites: Boston, MA; Chicago, IL; Detroit, MI area; Los Angeles, CA; Newark, NJ; and Pittsburgh, PA. A total of 3302 women between the ages of 42 and 52 years participated in this study from 1996 to 1997, which included 1550 whites, 935 blacks, 286 Hispanics, 250 Chinese, and 281 Japanese. Eligibility criteria for enrollment in SWAN included age 42–52 years, an intact uterus and at least one ovary, no use of exogenous hormones affecting ovarian function in the past 3 months, at least one menstrual period in the past 3 months, and self-identification with a designated racial/ethnic group at a site. The SWAN cohort has been followed up 16 times to date, and the baseline and the first 10 visits have been made public. Participants returned for regular examinations approximately annually. The institutional review board at each participating site approved the study protocol, and all participants provided written, signed informed consent.

Study protocols were approved by the Institutional Review Board at each site, and all participants provided written informed consent at each study visit. More details of the SWAN protocol have been published [20].

### 2.2. Inclusion and Exclusion Criteria

There were 3302 pre-menopause or perimenopausal women from SWAN 1996–1997 baseline survey. Firstly, 453 women period with hyperglycemia during pre-menopause were excluded. Remaining exclusion criteria included (1) received exogenous hormone therapy, (2) missing BMI value or BMI < 18.5 kg/m^2^, (3) missing or unknown menopause status, (4) no follow-up records after menopause, and (5) no sex hormones were measured at the follow-up visit. Hormone therapy was defined as (1) self-reported use of estrogen injections, and (2) self-reported use of any estrogen medication. After women confirmed natural menopause, the first follow-up was used to obtain measurements of sex hormones, and the last follow-up was used to determine hyperglycemia status. In all, 1438 pre-menopause or perimenopausal women were included in this study (Appendix A).

### 2.3. Covariates Measurements

Hormone analyses were performed in the SWAN Endocrine Laboratory at the University of Michigan using an ACS-180 automated analyzer (Bayer Diagnostics). FSH concentrations were determined using a 2-point chemiluminescent immunoassay, and SHBG and DHAS were determined using a competitive chemiluminescent assay. E2 concentrations were determined using a modified off-line ACS-180 (E2-6) immunoassay, and T concentrations were assessed using a modified ACS-180 total testosterone assay. Fasting serum glucose level was measured by the hexokinase method (Boehringer Mannheim Diagnostics, Indianapolis, IN, USA).

Common protocols were standardized and used by trained examiners across the seven sites. Information obtained by questionnaires included demographics (age, ethnicity, level of education, and so on), female physiology, medical history, and behavioral lifestyles. Anthropometric and laboratory data were collected by clinical technicians. Standing height and weight were measured in light clothing without shoes. BMI was calculated as weight in kilograms divided by height in meters squared. Menopause status was based on self-report of bleeding patterns. Perimenopause status includes early perimenopause and late perimenopause, and post-menopause status is defined as no bleeding for the past 12 months.

### 2.4. Definition of Exposure and Outcome

According to the BMI levels, participants were divided into the following groups, (1) normal: BMI < 25 kg/m^2^; (2) overweight/obesity: BMI ≥ 25 kg/m^2^. Women with incident diabetes were identified by the presence of one or more of the following conditions in any postmenopausal visit: (1) self-report of physician diagnosis of diabetes, (2) current use of insulin or glucose-lowering medication, and (3) a fasting glucose measurement >126 mg/dL. In addition, prediabetes was defined as impaired fasting glucose (110 mg/dL < fasting blood glucose < 126 mg/dL). Hyperglycemia was defined as individuals with prediabetes or diabetes.

### 2.5. Statistical Analysis

Figure 1 illustrates the two basic phases of this study, (1) extracting composite features of multiple sex hormones (SHBG, E2, FSH, T, DHAS) by a PLS regression method, and (2) examining the mediator role of PLS score and individual sex hormones through causal mediation analysis.

Generalized linear models were used to test the differences in study variables across normal-weight and overweight/obesity groups. Given the skewed distribution of SHBG, E2, T, FSH, and DHAS, logarithmic transformations with the natural base were applied. Logistic regression models were used to evaluate the association of overweight/obesity with hyperglycemia, diabetes, and prediabetes risk by adjusting for covariates of age, race, education level, physical activity, and smoking regularly.

PLS regression method is a multivariate dimensionality reduction method that generalizes the features of principal component analysis with those of multiple linear regression [21]. We applied the PLS regression to obtain the hormonal signature associated with BMI. Briefly, the PLS factor was retained as the linear combinations of predictor variables (i.e., E2, T, FSH, DHAS, and SHBG) in a PLS regression with BMI as a dependent variable. Each loading from the PLS regression quantified the contribution of each hormone to the PLS factor. The PLS score was calculated using a linear combination of the product of hormone loadings and levels.

Linear regression models were used to evaluate the association between overweight/obesity and mediators (PLS score and individual sex hormones). Logistic regression models were used to quantify the association between mediators and hyperglycemia risk with adjustment for overweight/obesity. These models were also adjusted for baseline age, race, ever smoking regularly, education level, physical activity, and follow-up years.

Finally, causal mediation analysis was used to evaluate the mediation effects of the PLS score and individual sex hormones on the associations of overweight/obesity with hyperglycemia. In the sensitivity analysis, causal mediation analysis was used to evaluate the mediation effects of the waist-related PLS score and individual sex hormones on the associations of abdominal obesity with hyperglycemia. The counterfactual causal mediation approach divides the total effect into natural indirect effects (NIE), which represent the partial effect of exposure affecting the outcome via the specified mediating variable, and natural direct effects (NDE), which represent the effect of exposure directly affecting the outcome. On account of the additivity of mediation effects on the excess relative risk (ERR) scale, the percentage of indirect effects was quantified using (NIE/(NIE + DIE)) × 100%. In our sensitivity analysis, we examined the mediating effects of overweight/obesity on diabetes and prediabetes, respectively.

All statistical tests were two-sided, and *p* < 0.05 was considered statistically significant. All data preprocessing and PLS regression analyses were performed using R software with the “pls” package (version 2.8-1) [22]. SAS software (Version 9.4, SAS Institute, Cary, NC, USA) was used to calculate the total effect, direct effect, and indirect effect through the proc “causalmed” procedure. The non-parametric method-based bootstrap, which was performed 1000 times, was used to estimate and examine the mediation effect [23].

## 3. Results

Table 1 summarizes the characteristics of the participants. At baseline, the mean age was 46.5 (standard deviation: 2.6) years. In total, 45.2% of women had a college degree or higher, and 41.7% were frequent smokers. At baseline and follow-up, there was no significant difference in age and follow-up duration. During a mean follow-up period of 9.9 years, 145 cases of hyperglycemia occurred, including 114 cases of incident diabetes and 31 cases of incident prediabetes. Significant differences were found for all sex hormones (SHBG, FSH, T, E2, DHAS) at follow-up. Compared with the normal-weight women, overweight/obese women had an increased risk of hyperglycemia, with adjusted odd ratios (ORs) of 4.06 and a 95% confidence interval (CI) of 2.52 to 6.80 (*p* < 0.001). The ORs were 3.81 (*p* < 0.001) and 2.79 (*p* = 0.013) for risk of diabetes and prediabetes, respectively (Appendix A).

Appendix A shows the correlation matrix of sex hormones, indicating that complex relations among sex hormones. FSH is negatively correlated with E2 and T (r = −0.39 and −0.1, all *p* < 0.05) and positively correlated with SHBG and DHAS (r = 0.16 and 0.06, all *p* < 0.05). T is positively correlated with E2 and DHAS (r = 0.21 and 0.35, all *p* < 0.05) and negatively correlated with SHBG (r = −0.11, *p* < 0.05). DHAS is negatively correlated with SHBG (r = −0.12, *p* < 0.05). The loadings obtained from PLS regression were −0.410, 0.528, 0.247, −0.728, and −0.123 for SHBG, E2, T, FSH, and DHAS, respectively. According to the root mean squared error of prediction, the first PLS component was selected (Appendix A). The PLS score is equal to the first PLS component. By using the first PLS component, we can explain 28.35% and 21.81% of the variation in the sex hormones and BMI level, respectively.

Figure 2a shows the standardized regression coefficients and 95% CIs of overweight/obesity on individual sex hormones and PLS score. After adjustments for confounders, compared to normal-weight women, the standardized regression coefficients (95% CIs) were −0.52 (−0.62 to −0.41), 0.38 (0.27 to 0.48), 0.26 (0.15 to 0.37), −0.52 (−0.63 to −0.42), and −0.13 (−0.23 to −0.02) for SHBG, E2, T, FSH, DHAS, respectively. Figure 2b shows the standardized ORs and 95% CIs of individual sex hormones and PLS score on hyperglycemia. The adjusted ORs (95% CIs) for the SHBG and FSH were 0.70 (0.58 to 0.84) and 0.69 (0.59 to 0.81). There was no statistically significant association between the levels of E2, T, DHAS, and hyperglycemia (*p* > 0.05).

Table 2 presents the causal mediation model parameters with overweight/obesity as the predictor, PLS score and each sex hormone as a single mediator, and hyperglycemia incidence as the outcome. The NIE of the PLS score was 1.24 (*p* < 0.001), indicating that being overweight/obese increases the risk of hyperglycemia by an additional 1.24-fold through altering the concentration of multiple sex hormones. Considering single sex hormone mediators, decreasing SHBG and FSH levels of overweight/obese women increases the risk of hyperglycemia incidence by 0.74-fold and 0.81-fold, respectively. E2, T, and DHAS all showed no statistically significant mediating effects (*p* > 0.05 for NIE and mediation proportion). The mediation proportions (95% CIs) of PLS score, SHBG, and FSH were 36.50% (23.15 to 49.86), 21.38% (10.33 to 32.43), 24.08% (13.77 to 34.38), respectively. Appendix A shows the causal mediation model parameters with prediabetes and diabetes as an outcome. Appendix A shows the causal mediation model parameters with abdominal obesity as exposure. All the results of mediation proportion in the sensitivity analysis are consistent.

## 4. Discussion

This study explored the mediation effect of sex hormones underlying the association between overweight/obesity and diabetes risk in menopausal women. We found that sex hormones were partial mediators in the relationship between overweight/obesity and hyperglycemia. A pooled score aggregating characters from multiple sex hormones mediated more than one-third of the overweight-hyperglycemia association, with positive correlations with T, E2 and negative correlations with SHBG, FSH, and DHAS. Both SHBG and FSH jointly mediate the associations between overweight/obesity and hyperglycemia.

Women are vulnerable to cardiovascular events during the perimenopausal period; therefore, rigorous self-management is necessary. Being overweight is an established risk factor for diabetes in perimenopausal women, but the underlying mechanisms remain unclear. The current study hypothesized that changes in hormone levels due to the accumulation of overweight/obesity may subsequently alter the risk of diabetes. The results showed that even though there were subtle differences in the subgroup analyses, both SHBG and FSH mediated the association between overweight and hyperglycemia to varying degrees.

Low SHBG levels seen in obesity are caused by the high lipid content of the liver and by high pro-inflammatory cytokines (TNF-α and IL-1) [24]. In addition, low SHBG is also closely associated with insulin resistance. Hepatic factors (proteins that are primarily secreted by the liver) are known to directly affect glucose metabolism [14,25,26], and SHBG may also be another important hepatic factor regulating glucose homeostasis [14,27]. A Finnish longitudinal study showed that BMI and SHBG were independent risk factors of abnormal glucose metabolism in midlife [28]; this is consistent with our findings. Moreover, some studies have found that diet habits, such as increased intake of rice, fish, and green vegetables, can effectively increase blood SHBG concentration independent of original body weight [29,30]. More information is needed in the future to identify strategies to increase circulating SHBG concentrations and to better inform possible prevention strategies for diabetes in overweight/obese women.

In females, FSH promotes the growth and maturation of ovarian follicles and estrogen production [31]. According to SWAN’s previous study, FSH levels followed three patterns over the menopausal transition: low, medium, and high FSH increases [32]. Obese and overweight women were less likely than lean women to fall into the high trajectory. Lower FSH levels due to high BMI level probably arise from feedback inhibitory effects of estrogen produced by aromatization in fat tissue on FSH production. A recent study conducted in East China in 1610 postmenopausal women not using hormone therapy (aged 55–89 years) reported that FSH was significantly inversely associated with fasting glucose and HbA1c [33]. FSH may also be associated with diabetes through inflammatory markers, such as c-reactive protein, TNF-α, and IL-1β [33]. Since FSH primarily exerts its biological function through binding to the FSH receptor (FSHR), it suggests FSHR may be a potential therapeutic target for diabetes control in overweight or obese postmenopausal women [34].

In addition to the separate effects of gonadotropins and binding proteins, there are also complex regulatory and transformation relationships between hormones, making it reasonable to use a dimensionality reduction approach to extract hormonal signaling features. Obese women tend to have higher free E2, higher estrone, and lower SHBG [35]. This is consistent with the composite hormone profile extracted in the current study. Even though E2 and FSH always change synchronously throughout the menopausal transition to post-menopause and have a significant negative association, no evidence was found in the current study that E2 is in the causal pathway of overweight/obesity and diabetes. A possible explanation is that E2 does not function as a circulating hormone in postmenopausal women due to its ovarian production is suppressed [36].

This is the first longitudinal study to examine the mediating role of sex hormones and binding proteins in overweight-diabetes risk in women during the menopausal transition period, providing a new perspective to unravel the complex regulatory mechanisms of diabetes. Considering the complex correlations among hormones, we use a composite score to extract multiple hormonal signals that go together to explain the increased risk of diabetes due to overweight/obesity. There are also some limitations in the current study. Although PLS regression has been used to extract complex correlation features, further studies are needed to elucidate the relationship between sex hormones and SHBG and diabetes risk because the relationship between sex hormones and SHBG is highly dynamic and complex. Further, the presence of unknown confounding bias is inevitable despite adjusting for several known confounders. Another limitation could have been that self-reported and diagnostic information in the SWAN cohort does not distinguish between type 1 and type 2 diabetes. However, almost all of the diabetes cases in this life stage can be assumed to be type 2 diabetes. Furthermore, due to the limited information on medications and treatments in the questionnaire, the effect of medications cannot be completely ruled out. Finally, the lack of HBA1c measurements may lead to an inaccurate definition of diabetes or prediabetes. Future studies using biomarkers reflecting inflammatory states, impaired insulin signaling, and hyperinsulinemia are needed to fully assess the role of these pathways in explaining the effects of overweight/obesity on diabetes.

## 5. Conclusions

In conclusion, sex hormones mediated the effects of overweight/obesity on diabetes, especially SHBG and FSH. The quantification of the multiple mediation effects will help clinicians better understand the underlying mechanistic process of overweight/obesity, sex hormones, and diabetes in menopausal women. Specifically, in addition to the necessary weight control, sex hormone surveillance and supplementation therapy are recommended for women who are above normal weight to reduce diabetes risk during the menopause period.

## Figures and Tables

**Figure 1 ijerph-20-01831-f001:**
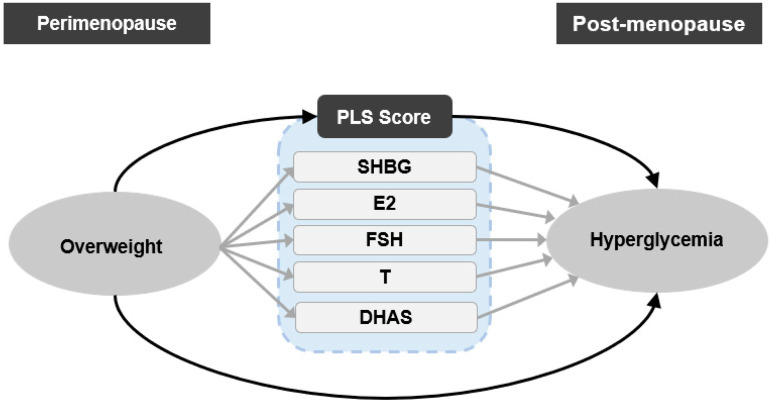
Scheme of the study design. Single-headed arrows represent regression paths; ovals represent exposure and outcome variables, and rounded rectangles represent mediator variables. PLS, partial least squares; SHBG, sex hormone binding protein; E2, estradiol; T, testosterone; FSH, follicle stimulating hormone; DHAS, Dehydroepiandrosterone sulfate.

**Figure 2 ijerph-20-01831-f002:**
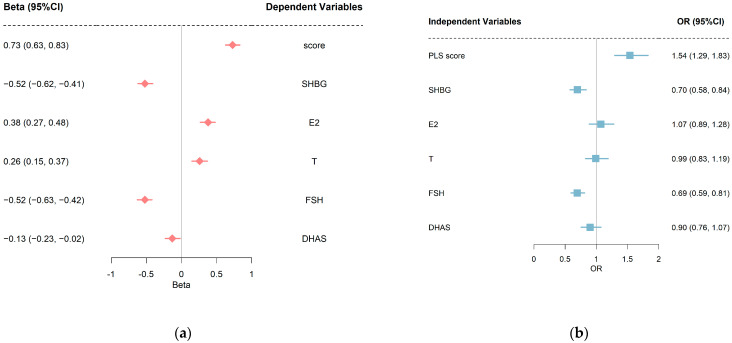
(**a**) Standard regression coefficients and 95% CIs of overweight/obesity on sex hormones. (**b**) Standard OR coefficients and 95% CIs of sex hormones on hyperglycemia. Covariates adjusted in linear regression models were baseline age, race, smoking regularly, education, physical activities, and follow-up years. Covariates adjusted in logistic regression models were baseline age, race, smoking regularly, education, physical activities, follow-up years, and overweight/obesity. PLS, partial least squares; SHBG, sex hormone binding protein; E2, estradiol; T, testosterone; FSH, follicle-stimulating hormone; DHAS, Dehydroepiandrosterone sulfate.

**Table 1 ijerph-20-01831-t001:** Characteristics of participants at baseline and follow-up.

	Normal Weight (*n* = 662)	Overweight/Obesity (*n* = 776)	Total (*n* = 1438)	*p*
**Baseline**				
Age, years	46.4 (2.6)	46.5 (2.7)	46.5 (2.6)	0.793
FBG, mg/dL	88 [84, 94]	92 [87, 98]	90 [86, 96]	<0.001
BMI, kg/m^2^	22.1 (1.7)	31.6 (6)	27.3 (6.6)	<0.001
Waist, cm	73 (5.7)	93.1 (12.9)	83.8 (14.3)	<0.001
Frequent smoker, *n* (%)	248 (37.7)	347 (45.1)	595 (41.7)	0.006
Race				<0.001
White	326 (49.2)	374 (48.2)	700 (48.7)	
Japanese	111 (16.8)	30 (3.9)	141 (9.8)	
Chinese	127 (19.2)	29 (3.7)	156 (10.8)	
Black	79 (11.9)	292 (37.6)	371 (25.8)	
Hispanic	19 (2.9)	51 (6.6)	70 (4.9)	
Education				<0.001
Less than High School	32 (4.8)	59 (7.6)	91 (6.3)	
High School Graduate	106 (16.0)	131 (16.9)	237 (16.5)	
Some College /Technical School	187 (28.2)	273 (35.2)	460 (32.0)	
College Graduate	165 (24.9)	133 (17.1)	298 (20.7)	
Post Graduate Education	172 (26.0)	180 (23.2)	352 (24.5)	
Physical activity				<0.001
Much less than other women	17 (2.6)	43 (5.5)	60 (4.2)	
Somewhat less than other women	43 (6.5)	161 (20.7)	204 (14.2)	
About the same as other women	265 (40.0)	323 (41.6)	588 (40.9)	
Somewhat more than other women	206 (31.1)	174 (22.4)	380 (26.4)	
Much more than other women	131 (19.8)	75 (9.7)	206 (14.3)	
**First follow-up**				
Age, years	53.1 (2.7)	52.9 (2.9)	53.0 (2.8)	0.386
Follow-up, years	6.6 (2.4)	6.5 (2.5)	6.5 (2.5)	0.212
SHBG, nM	49 [33.8, 66.5]	36.2 [26.0, 50.3]	41.2 [28.5, 58.4]	<0.001
T, ng/dL	34.3 [24.1, 48.0]	40 [29.0, 54.3]	37.1 [26.3, 51.1]	<0.001
FSH, mIU/mL	116.5 [90.8, 143.9]	87.8 [62.3, 118.8]	100.3 [75.3, 132.1]	<0.001
E2, pg/mL	14 [10.3, 19.2]	18.7 [13.2, 26.6]	16.2 [11.8, 23.4]	<0.001
DHAS, µg/dL	118.5 [81.2, 175.1]	109.7 [71.2, 159.3]	113.8 [75.5, 167.7]	0.004
**Last follow-up**				
Age, years	56.4 (2.7)	56.3 (2.8)	56.3 (2.8)	0.651
Follow-up, years	9.9 (0.9)	9.8 (1.4)	9.9 (1.2)	0.088
Hyperglycemia, *n* (%)	25 (3.8)	120 (15.5)	145 (10.1)	<0.001
Diabetes mellitus, *n* (%)	17 (2.6)	97 (12.5)	114 (7.9)	<0.001
Prediabetes, *n* (%)	8 (1.2)	23(3.0)	31(2.2)	<0.001

Variables were described as mean (standard deviation), median [interquartile range], and *n* (%). BMI, Body Mass Index; SHBG, Sex Hormone Binding Globulin; T, Testosterone; E2, Estradiol; FSH, Follicle-stimulating hormone; DHAS, Dehydroepiandrosterone sulfate; FBG, fasting blood glucose.

**Table 2 ijerph-20-01831-t002:** The Mediation effect of PLS score and each sex hormone on association between overweight/obesity and hyperglycemia by excess relative risk (ERR) scale.

	NIE (95% CI)	NDE (95% CI)	TE (95% CI)	Mediation Proportion %
PLS score	**1.24 (0.51, 1.97) ****	2.14 (0.64, 3.65) *	3.38 (1.35, 5.42) *	**36.50 (23.15, 49.86) ****
SHBG	**0.74 (0.22, 1.27) ***	2.72 (0.99, 4.45) *	3.46 (1.40, 5.52) *	**21.38 (10.33, 32.43) ****
FSH	**0.81 (0.29, 1.32) ***	2.55 (0.88, 4.22) *	3.36 (1.34, 5.38) *	**24.08 (13.77, 34.38) ****
T	−0.02 (−0.19, 0.15)	3.37 (1.36, 5.39) *	3.35 (1.36, 5.34) *	−0.83 (−7.11, 5.45)
E2	0.14 (−0.17, 0.45)	3.21 (1.26, 5.17) *	3.35 (1.36, 5.35) *	4.10 (−5.62, 13.09)
DHAS	0.07 (−0.04, 0.19)	3.29 (1.32, 5.25) *	3.36 (1.36, 5.36) *	2.13 (−1.28, 5.55)

SHBG, Sex Hormone Binding Globulin; T, Testosterone; E2, Estradiol; FSH, Follicle-stimulating hormone; DHAS, Dehydroepiandrosterone sulfate; PLS, Partial least squares analysis; NDE, Natural direct effect; NIE, Natural indirect effect; TE, Total effect. Mediation models were adjusted for baseline age, race/ethnicity, smoking regularly, education, physical activities, and follow-up years. * *p* < 0.05, ** *p* < 0.001. All coefficients were standardized by z-transformation before analysis.

## Data Availability

Data are public and the data sets analyzed in the current study are available on the web: https://www.nia.nih.gov/research/dgcg/study-womens-health-across-nation-swan-repository (accessed on 5 March 2019).

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
