# Peer review of "Association between Body Mass Index and Diabetes Mellitus Are Mediated through Endogenous Serum Sex Hormones among Menopause Transition Women: A Longitudinal Cohort Study"

_ijerph, 2023, doi:10.3390/ijerph20031831_

Round 1

Reviewer 1 Report

The authors systemically examined the mediating effects of six sex hormones in the association between obesity and diabetes risks, leveraging a relatively large sample of pre-menopause or perimenopausal women. The findings are interesting and of relevance. This paper is well written. Some minor comments may help authors improve their manuscript.

1. The study design should have to be presented more clearly in the title or abstract.

2. The methods of follow-up were missing. How was incidence of diabetes identified?

3. Could sex hormones affect obesity? Or, can you examine the temporal association between sex hormones and obesity?

4. Only fasting glucose was used to define diabetes. This may result in missclassification of diabetes. 

Reviewer 2 Report

It is a real pleasure to review this manuscript. In this work, the authors carried out a mediation analysis to study whether endogenous serum sex hormones mediated the association between body mass index and type II diabetes (T2D). I have several comments on this work.

 1. As the onset time can be available, the survival models such as Cox model can be used for the b arm.

2. Did there exist the re-use of data between the mediation analysis and PLS analysis?

3. As SHBG, E2, FSH, T and DHAS showed complex relations; it was not clear that the discovered mediation associations were driven by a single or multiple hormones. Thus, it may be better to include them together into the mediation model, at least into the mediator-outcome model.

4. It was not clear how to calculate the contribution of indirect effect, using a log or non-log scale? In addition, parentheses were missed in line 159.

5. Moreover, because both linear and non-linear models were used, the relation of TE= NIE+NDE often did not hold under this case. It needed to highlight and explain this point.

Reviewer 3 Report

Do the correlations between sex hormones is an unexpecting finding? In my opinion, No, therefore, it should be underlined in both Result and Discussion sections.

The finding “we selected the first  PLS factor (Figure S3), which explained 28.39% and 22.38% of the variance in sex hormones and BMI level, respectively” must be clearly explained in the Result section. In my opinion, it can be hard for many readers to understand the meaning of this finding.

Sex-hormone supplementation therapy is a complex issue and the presented data do not give an evidence that sex-hormone supplementation is the best choice to fight against diabetes. Therefore, it is hard to accept that “sex hormone surveillance and supplementation therapy are also needed for women who are above normal weight to reduce diabetes risk during the menopause period”

Reviewer 4 Report

In the manuscript, “Association between body mass index and diabetes mellitus are mediated through endogenous serum sex hormones among menopause transition women” the authors investigate an interesting relationship between overweight and/or obesity and hyperglycemia mediated through sex hormones testosterone, estradiol, FSH and DHAS and the globulin sex hormone binding globulin. The results are important to understand the involvement of the unique endocrine milieu in menopausal women, which increases their risk for cardiometabolic disease.

There are some aspects to consider:

Abstract

The abstract online and the abstract in the type set document have some discrepancies – please revise.

Introduction and Aim

Have the authors considered the effect of cortisol on this transition (Woods et al, 2009: https://www.ncbi.nlm.nih.gov/pmc/articles/PMC2749064/) as this may also mediate the effect of the hormones under investigation (especially DHAS).

Methods

Have the authors information on the cardiometabolic health of the women at baseline and at follow-up? Prevalence of hypertension, stroke risk, dyslipidemia etc? What was the lipid profile of these women at baseline and follow-up? The inflammatory profile at baseline and follow-up should be reported and the effect of chosen markers (for example CRP or TNF-alpha) should be assessed within the PLS model. Additionally, the use of any anti-hypertensive and lipid medication should be reported and the effect assessed on the PLS model.

What was the motivation behind using BMI instead of waist circumference (linking better to insulin resistance)? Has the effect of waist circumference been investigated within this context and PLS model?

Have multiple PLS models been considered for overweight and obese groups seperately? This might be an important consideration given the sex hormones chosen for the current PLS model.

Discussion

The discussion is well written and the authors have taken particular care to effectively and meaningfully interpret the results. However, the authors should interpret their results using the relevant literature – thus explain why and how their results relate or disprove current understanding.

Round 2

Reviewer 2 Report

thanks for the responses and I have no more comments.